# Large Pre-trained time series models for cross-domain Time series analysis tasks

**Harshavardhan Kamarthi**
College of Computing
Georgia Institute of Technology
`harsha.pk@gatech.edu`

**B. Aditya Prakash**
College of Computing
Georgia Institute of Technology
`badityap@cc.gatech.edu`

## Abstract

Large pre-trained models have been vital in recent advancements in domains like language and vision, making model training for individual downstream tasks more efficient and provide superior performance. However, tackling time-series analysis tasks usually involves designing and training a separate model from scratch leveraging training data and domain expertise specific to the task. We tackle a significant challenge for pre-training a foundational time-series model from multi-domain time-series datasets: extracting semantically useful tokenized inputs to the model across heterogenous time-series from different domains. We propose Large Pre-trained Time-series Models (LPTM) that introduces a novel method of *adaptive segmentation* that automatically identifies optimal dataset-specific segmentation strategy during pre-training. This enables LPTM to perform similar to or better than domain-specific state-of-art model when fine-tuned to different downstream time-series analysis tasks and under zero-shot settings. LPTM achieves superior forecasting and time-series classification results taking up to 40% less data and 50% less training time compared to state-of-art baselines.

## 1 Introduction

Time-series analysis tasks are important well-studied problems such as forecasting (Hyndman and Athanasopoulos, 2018) and classification (Chowdhury et al., 2022) with applications in wide-ranging domains such as retail, meteorology, economics, and health. Recent works (Chen et al., 2021; Wang et al., 2022; Zeng et al., 2023) have shown the efficacy of purely data-driven deep learning models. However, most state-of-art neural models are known to be data-hungry and require substantial training data. Motivated by language and vision foundational models Bommasani et al. (2021), recent body of works build pre-trained time-series models Das et al. (2023b); Ansari et al. (2024); Jin et al. (2024); Rasul et al. (2023). These models are trained on diverse datasets from different domains during pre-training. They require less training resources and data and produce superior performance. These models can also be deployed without any training, in a zero-shot or few-shot setting.

These foundational models, however, require large amounts of data for pre-training, which is still a challenge for time-series datasets. Moreover, they do not provide consistent performance improvement across all the domains. We identify an important challenge to building a unified pre-trained model for time-series that is pre-trained on and deployed to multiple domains: representation of diverse time-series input into these models.

Most neural sequential models input time-series values for each time-step separately. However, unlike text data, each individual time stamp may not provide enough semantic meaning about local temporal patterns of the time series. To tackle this, Nie et al. (2022) proposed to segment the time series and input each segment as individual tokens to their transformer-based model. This simple segmentation method of *tokenizing* time-series has been used by recent pre-trained models Woo et al.

38th Conference on Neural Information Processing Systems (NeurIPS 2024).

(2024); Das et al. (2023b) to provide superior performance across multiple applications. However, segmenting input time-series uniformly with fixed-length segments, while simple, can be a very inflexible tokenizing method, especially when dealing with datasets from multiple domains with different set of underlying generative dynamics, sampling rate, noise, etc. For example, among two datasets, a dataset with a lower sampling rate (such as GDP time-series) may require longer segments than those with higher sampling rates to capture similar patterns in the model (such as heart sensors collecting data in milliseconds). However, note that the dynamics of the same time-series may vary with time Liu et al. (2024a). For, time intervals that are smoother with less complex dynamics, using longer segment sizes may suffice but intervals where time-series are complex and have multiple temporal patterns may require finer-grained segmentation. For example, for seasonal epidemics, time-series is smoother during the off-season and has more complex dynamics during outbreaks and times of higher incidence.

We tackle this important problem of representing diverse time-series datasets when training a pre-trained foundational model for time-series and propose **Large Pre-trained Time-series Models** (LPTM), a novel method for generating pre-trained models for time-series data across multiple domains. LPTM uses a simple transformer-based architecture and leverages masking-based self-supervised pre-training to train on multiple datasets from different domains. Our main contribution focuses on how we input time-series segments as tokens to the transformer. To overcome the challenges associated with segmentation on diverse datasets discussed above, we propose a novel *adaptive segmentation module* that segments the time-series of each domain based on how well it performs on self-supervised pre-training. The segmentation module uses a novel scoring mechanism during pre-training to identify an effective segmentation strategy for a domain. LPTM can be fine-tuned or deployed in a zero-shot setting to various forecasting and classification tasks in domains such as epidemiology, energy, economics, behavioural datasets, etc. LPTM provides performance on par with state-of-art models with lesser pre-training data, training data and fewer training steps. Our main contributions can be summarized as follows:

• **Multi-domain Pre-trained time-series model** We propose a framework for generating foundational pre-trained models for time-series that are trained on multiple datasets across varied domains. LPTM solves the tokenization problem for cross-domain time-series data and proposes a novel adaptive segmentation module which is important to build pre-trained models for time-series similar to foundational models for text and vision.

• **Adaptive segmentation for cross-domain pre-training** To optimally extract semantically useful information from time-series of different domains with varied dynamics and sampling rates for pre-training, we propose a novel adaptive segmentation module that learns segmentation strategy for each domain based on losses from self-supervised learning tasks.

• **State-of-art and efficient performance in diverse downstream time-series tasks** We evaluate LPTM on downstream forecasting and classification tasks from multiple domains and observe that LPTM consistently provides performance similar to or better than previous state-of-art models usually under zero-shot evaluation as well as when fine-tuned with lesser training data and compute time. Overall, we also observe that LPTM typically requires less than 80% of training data used by state-of-art baselines to provide similar or better performance.

## 2 Problem Setup

**Time-series analysis tasks**  Our pre-trained model can be used for many time-series tasks including forecasting and classification from multiple benchmarks and domains. For a given downstream task let $\mathcal{D}^T$ be the time-series dataset consisting of time series $\mathbf{y}^{1\ldots T}$. A time-series analysis task's goal is to predict important properties of the time-series. For example, the forecasting task involves predicting the future values $\mathbf{y}^{T+1\ldots T+K}$ whereas classification involves predicting the class label of the input time-series based on labelled training data.

**Self-supervised pre-training on multi-domain datasets**  The goal of our work is to learn useful knowledge and patterns from time-series datasets from time-series from different domains. Formally, we have access to time-series datasets from $K$ domains where the datasets of domain $k$ is denoted as $\mathcal{D}'_k = \{\mathcal{D}'_{k,i}\}_{i=1}^{N(k)}$ where $N(k)$ is the number of datasets in domain $k$. Examples of these domains include epidemiology, energy forecasting, macroeconomics, traffic prediction, etc. The entire set of

heterogenous multi-domain *pre-train* dataset is denoted as $\mathcal{D}_{\text{pre}} = \{\mathcal{D}'_1, \mathcal{D}'_2, \ldots, \mathcal{D}'_K\}$. In order to effectively pre-train LPTM on $\mathcal{D}_{\text{pre}}$ we formulate the problem as a set of self-supervised learning tasks $\mathcal{T}_{\text{pre}} = \{\mathcal{T}_i\}_{i=1}^R$ on the set of pre-training datasets $\mathcal{D}_{\text{pre}}$. During pre-training, we sample $(\mathcal{D}'_{k,i}, k)$, a dataset and its domain label from $\mathcal{D}_{\text{pre}}$ and train the model on each of the self-supervised learning tasks in $\mathcal{T}_{\text{pre}}$. The tasks in $\mathcal{T}_{\text{pre}}$ are self-supervised and do not require additional labels or other ground truth. These tasks mask patches of the input data and train the model to recover the original input.

Therefore, our problem can be formally stated as: *Given a heterogeneous set of multi-domain datasets $\mathcal{D}_{pre}$ and their domain labels, we train a model leveraging SSL tasks $\mathcal{T}_{pre}$ that learns important patterns and knowledge that can be leveraged on fine-tuning the model to any time-series analysis task on any novel dataset from any of the domains $d \in \{1, 2, \ldots, K\}$.*

## 3 Methodology

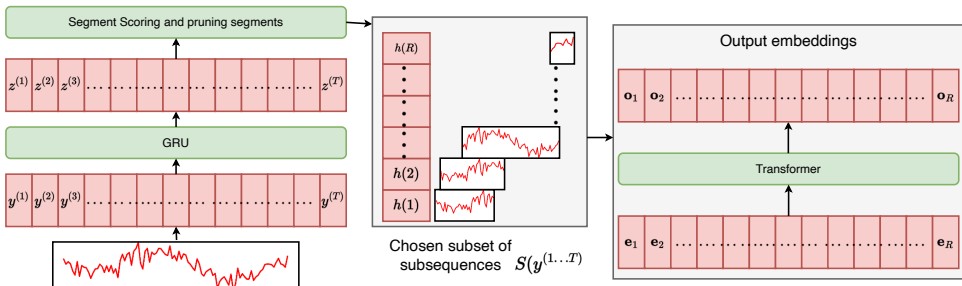

Figure 1: Overview of LPTM. The input time-series $y^{(1\ldots T)}$ is first segmented based on a scoring function optimized using SSL loss. The segments are fed as individual tokens to the transformer encoder to get output embeddings of time-series that are used for downstream tasks.

Similar to model pipelines used in NLP and vision, we first train a pre-trained model $M(\theta_{pre})$ on multiple pre-training datasets $\mathcal{D}_{\text{pre}}$. Most parameters of the pre-trained model $\theta_{pre}$ are trained over all the datasets and tasks. However, we use a separate segmentation module for each dataset domain to capture varied sizes of segments that differ across datasets. These segments are used as tokens for a transformer model that shares the parameters across all the tasks. For each pre-trained and downstream task, we append a final linear layer on the output embeddings of the transformer to generate the final prediction. Note that during fine-tuning on downstream tasks we update the parameters of all the modules of LPTM.

### 3.1 Adaptive Segmentation module

Previous works input each time-step of a time-series as individual tokens or fixed-sized segments. Unlike text, individual time-steps do not typically provide any semantic meaning about the temporal patterns of the time-series. Therefore, Nie et al. (2022) proposed to segment the input time-series into uniform length segments and use each of the segments as tokens to the transformer model. Segments of fixed size are also too inflexible to capture semantics of sequences that show varied behaviour across time and across datasets. Different pre-trained datasets may have varied temporal scales, periodicity and other temporal dynamics that cannot be encompassed by a single uniform segmentation strategy. For example, epidemic time-series are usually observed weekly and may have characteristic properties like seasonality, peaks and sudden outbreaks that should be captured by segmentation. Economic time-series, in contrast, are typically captured every quarter and are more monotone with sudden anomalies and changes in data distribution. Moreover, using a uniform segmentation may not be ideal for time series with multi-scale trends with some time-stamps having denser temporal information requiring finer-graned segmentation than others. Therefore, our goal is to identify an independent segmentation strategy for each domain of time-series dataset.

**Overview** LPTM automatically identifies an effective segmentation strategy for each dataset domain during pre-training. The overarching goal of the segmentation module involves identifying segments that lead to lower SSL loss. The segmentation module first scores all possible segments of

the input time-series and the segments with the highest scores are then chosen as the output segments used to tokenise the time-series. The scoring function is trained such that the score of the segments will reflect how likely the chosen segments will lower the SSL loss.

**Details** For a given input time-series $\mathbf{y}^{(1...t)}$, we pass it through a GRU to get hidden embeddings $\{\mathbf{z}^{(i)}\}_{i=1}^t$ that models the temporal patterns of the input:

$$\{\mathbf{z}^{(i)}\}_{i=1}^t = \text{GRU}_1(\{y^{(i)}\}_{i=1}^t). \tag{1}$$

We then introduce a *segment score function s* that provides a scalar score for any subsequence of the input time-series:

$$s(i,j) = \mathbf{v}^T \tanh\left(\mathbf{W}_1 \mathbf{z}_i + \mathbf{W}_2 \mathbf{z}_j + \mathbf{b}\right). \tag{2}$$

Intuitively, the score $s(i,j)$ for a subsequence from time-stamp $i$ to $j$ denotes how good the given segment is for the dataset when optimizing for the SSL loss.

In next step, we sample subset $S(y^{(1...t)})$ of subsequences over the time-series that a) covers the entire input time-series, b) has a high score function value. While retrieving the optimal $S(y^{(1...t)})$ is an interesting combinatorial optimization problem, we generate $S(y^{(1...t)})$ using a simple process as follows: for each $i \in \{1, 2, \ldots, t-1\}$, we denote $h(i) = \arg\max_{j \in \{i+1...,t-1\}} s(i,j)$ as the best segment starting from time-step $i$. Then we generate the set of segments $\hat{S}(y^{(1...t)}) = \{(i, h(i))\}_{i=1}^{t-1}$. To reduce the number of segments, we iteratively remove the lowest-scoring segments until we cannot remove any more segments without having time-steps not covered by any segments in the set. The final set of segments after pruning is denoted as $S(y^{(1...t)})$. The segmentation procedure is summarized in Alg. 1.

---

**Algorithm 1:** Adaptive Segmentation Module

---

   **Input** : Time-series $\mathbf{y}^{(1...t)} = \{y^{(i)}\}_{i=1}^t$
1  **Procedure** GetScores($\mathbf{y}^{(1...t)}$)
2     $\{\mathbf{z}^{(i)}\}_{i=1}^t \leftarrow \text{GRU}(\{y^{(i)}\}_{i=1}^t)$ ;            // Encode time-series with GRU
3     **for** $i \in \{1, 2, \ldots, t-1\}$ **do**
4         **for** $j \in \{i+1, \ldots, t\}$ **do**
5            $s(i,j) \leftarrow \mathbf{v}^T \tanh\left(\mathbf{W}_1 \mathbf{z}^{(i)} + \mathbf{W}_2 \mathbf{z}^{(j)} + \mathbf{b}\right)$ ; // Scores for all segments
6     **return** $\{s(i,j)\}_{i,j \in \{1,\ldots,t\}}^{i<j}$;

7  **Procedure** ChooseSegments($\{s(i,j)\}_{i,j \in \{1,\ldots,t\}}^{i<j}$)
8     **for** $i \in \{1, 2, \ldots, t-1\}$ **do**
9         $h(i) \leftarrow \arg\max_{j \in \{i+1,\ldots,t\}} s(i,j)$;     // Best segment starting at index $i$
10    $\hat{S} \leftarrow \{(i, h(i))\}_{i=1}^{t-1}$;
11    $i \leftarrow \arg\min_{j:(j,h(j)) \in \hat{S}} h(j)$;          // Select lowest scoring segment
12    **while** *Segments in $\hat{S} - \{(i, h(i))\}$ doesn't cover all time-steps $\{1, 2, \ldots, t\}$* **do**
13         $\hat{S} \leftarrow \hat{S} - \{(i, h(i))\}$;
14         $i \leftarrow \arg\min_{j:(j,h(j)) \in \hat{S}} h(j)$;
15    **return** $S \leftarrow \hat{S}$;

---

To generate the token embeddings $\hat{e}(i,j)$ for each segment $(i,j)$, we pass the embeddings $\{\mathbf{z}^{(i)}, \mathbf{z}^{(i+1)}, \ldots, \mathbf{z}^{(j)}\}$ through a self-attention layer used in transformers and aggregate the output embeddings.

$$\hat{e}(i,j) = \sum_{k=i}^j \text{Self-Atten}\{\mathbf{z}^{(i)}, \mathbf{z}^{(i+1)}, \ldots, \mathbf{z}^{(j)}\} \tag{3}$$

Additionally, we concatenate the following features to the token embedding of each segment token to provide information regarding the position and length of the segment: (1) Positional encoding of the starting time-step of the segment $pos(i)$ defined as:

$$pos(i) = \begin{cases} \sin(i/10^{5i/D}) & \text{if } i \text{ is even} \\ \cos(i/10^{5(i-1)/D}) & \text{if } i \text{ is odd}. \end{cases} \tag{4}$$

where $D$ is the dimensions of output embedding of self-attention over $\{\hat{\mathbf{e}}_i, \hat{\mathbf{e}}_{i+1}, \ldots, \hat{\mathbf{e}}_R\}$. (2) Positional encoding of the length of the segment $pos(j - i)$. The final output of the segmentation module is a sequence $\{\mathbf{e}_i\}_{i=1}^{R}$ where $R$ number of segments. The segments are arranged based on the ascending order of the first time-stamp of each segment. The token embeddings are fed into a stack of transformer layers similar to the encoder of the transformer Vaswani et al. (2017). The output of the transformer layers is a sequence of output embeddings denoted as $\{\mathbf{o}_i\}_{i=1}^{R}$.

## 3.2 Self-supervised learning Tasks

Pre-training on a wide range of heterogeneous datasets from multiple domains helps LPTM learn from useful patterns and latent knowledge across these domains that can be generalized to range downstream tasks on different domains. We propose two general self-supervised learning tasks motivated by pre-trained language models to enable LPTM to learn from all pre-trained datasets. We leverage a transformer model and use the segment token embeddings of the segmentation module. The two pre-training SSL tasks are **Random Masking** (RANDMASK) and **Last token masking** (LASTMASK). RANDMASK allows the model to extrapolate and interpolate masked segments of the input time-series. RANDMASK has also been explored for representation learning in previous works (Zerveas et al., 2021; Nie et al., 2022) but are applied on the same dataset as that used for training unlike our data and task-agnostic pre-training setup. Formally, we mask each input segment token with a probability of $\gamma$ and decode the values of time-series of the masked segments from the output embeddings of the transformer. We use a simple GRU with a single hidden layer on the output embedding of the masked token to decode the values of the segment. We use mean-squared error as the loss $\mathcal{L}_{SSL}$. LASTMASK is similar to RANDMASK except we mask last $\gamma$ fraction of the segments. This allows the model to forecast the future values of the time-series, an important task in many time-series domains.

## 3.3 Training details

**Instance normalization**   The values of the time-series of each dataset can vary widely based on the time-series domain. Therefore, as part of pre-processing, we first normalize the time-series of each dataset of pre-train datasets independently. Moreover, the data distribution and the magnitude of the time-series can vary across time. We use reversible instance normalization (REVIN) layer Kim et al. (2021). REVIN performs instance normalization on the input time-series and reverses the normalization of the output values. The normalization step is part of the neural model and gradients are calculated over the normalization and reverse normalization layers.

**Training the score function**   We use the loss from the SSL tasks to train the score function of the segmentation module and GRU in Eqn. 1. Since there is no direct gradient flow between the score function and the final predictions, due to the discrete nature of choosing the segments, we match the aggregated scores of all the chosen segments in $S(y^{(1 \ldots t)})$ to the negative logarithm of the total MSE loss of both SSL tasks:

$$\mathcal{L}_g = \left( \sum_{(i,j) \in S(y^{(1 \ldots t)})} s(i,j) + \log(\mathcal{L}_{SSL}) \right)^2 \tag{5}$$

where $\mathcal{L}_{SSL}$ is the total loss of both SSL tasks. We also backpropagate over $\mathcal{L}_g$ once every 10 batches to stabilize training since changing the segmentation strategy for every batch leads to unstable and inefficient training.

**Linear-probing and fine-tuning**   Kumar et al. (2022) showed that fine-tuning all the parameters of the pre-trained model for a specific downstream task can perform worse than just fine-tuning only the last layer (linear probing), especially for downstream tasks that are out-of-distribution to pre-trained data. To alleviate this, they suggest performing a two-stage fine-tuning process: we first perform linear probing followed by fine-tuning all the parameters.

# 4   Related Works

**Neural models for time-series analysis**   DeepAR Salinas et al. (2020) is a popular forecasting model that trains an auto-regressive recurrent network to predict the parameters of the forecast

distributions. Deep Markov models Krishnan et al. (2017); Rangapuram et al. (2018); Li et al. (2021); Gu et al. (2021) model the transition and emission components with neural networks. Recent works have also shown the efficacy of transformer-based models on general time-series forecasting Oreshkin et al. (2019); Zhou et al. (2021); Chen et al. (2021); Zhou et al. (2022); Liu et al. (2021). However, these methods do not perform pre-training and are trained independently for each application domain. therefore, they do not leverage cross-domain datasets to generate generalized models that can be used for a wide range of benchmarks and tasks.

**Self-supervised learning for time-series**    Recent works have shown the efficacy of self-supervised representation learning for time-series for various classification and forecasting tasks in a wide range of applications such as modeling behavioral datasets Merrill and Althoff (2022); Chowdhury et al. (2022), power generation Zhang et al. (2019), health care Zhang et al. (2022). Franceschi et al. (2019) used triplet loss to discriminate segments of the same time-series from others. These works use methods such as contrastive losses Eldele et al. (2021); Yue et al. (2022) or other similarity metric-based techniques Tonekaboni et al. (2021). However, all these methods apply SSL on the same dataset that is used for training and may not adapt well to multi-domain datasets. There has been some recent works leveraging foundational models like LLMs for time-series forecasting across multiple applications. One set of works directly uses LLMs without fine-tuning to perform time-series forecasting via careful prompting Gruver et al. (2024); Jin et al. (2024); Liu et al. (2024d). Other works fine-tune LLMs specifically for time-series forecasting Zhou et al. (2023); Rasul et al. (2023); Ansari et al. (2024). The time-series representation used by these models includes using individual time-steps as input Ansari et al. (2024), converting each digit of the time-series to character embeddings to be directly used by LLMs Gruver et al. (2024); Jin et al. (2024) and uniform segmentation Das et al. (2023b); Woo et al. (2024); Zhou et al. (2023). LPTM performs superior to these methods while being 10x to 100x smaller than the large LLMs used as backbones.

# 5   Experiment Setup

**Datasets**    We derive pre-train time-series datasets from multiple domains: • **Epidemics:** We use a large number of epidemic time-series aggregated by Project Tycho (van Panhuis et al., 2018). from 1888 to 2021 for different diseases collected at state and city levels in the US. We remove time series with missing data and use time series for 11 diseases of very diverse epidemic dynamics such as seasonality, biology, geography, etc.: Hepatitis A, measles, mumps, pertussis, polio, rubella, smallpox, diphtheria, influenza, typhoid and Cryptosporidiosis (Crypto.). • **Electricity:** We use ETT electricity datasets (ETT1 and ETT2) collected from (Zhou et al., 2021) at 1 hour intervals over 2 years. We use the default 12/4/4 train/val/test split and use the train split for pre-training as well. • **Traffic Datasets:** We use 2 datasets related to traffic speed prediction. PEMS-Bays (PEM-B) and METR-LA (Li et al., 2017) are datasets of traffic speed at various spots collected by the Los Angeles Metropolitan Transportation Authority and California Transportation Agencies over 4-5 months. • **Demand Datasets:** We use bike and taxi demand datasets (NY-B, NY-T) from New York City collected from April to June 2016 sampled every 30 minutes. We all but the last 5 days of data for training and pre-training. • **Stock forecasting**: We also collect the time-series of daily stock prices of Nasdaq and S&P 500 index using Yahoo finance data (Xu and Berkely, 2014) from July 2014 to June 2019. We train and pre-train using the first 800 trading days and use the last 400 for testing. • **M4 competition time-series**: We also used the 3003 time-series of M4 forecasting competition (Makridakis and Hibon, 2000) which contains time-series from multiple domains including demographics, finance, and macroeconomics. • **Motion and behavioral sensor datasets**: We use the set of sensor datasets extracted from UEA archive (Bagnall et al., 2018) and UCI Machine learning repository (Asuncion and Newman, 2007) similar to (Chowdhury et al., 2022). Note that our publicly accessible pre-training dataset is significantly smaller than other pre-trained datasets used by past work Ansari et al. (2024); Das et al. (2023b) some of which use confidential data inaccessible to us. We also do not use any synthetic datasets like Das et al. (2023b); Ansari et al. (2024).

**Downstream tasks**    We test the pre-trained LPTM trained on datasets discussed above on multiple forecasting and time-series classification tasks. We perform forecasting on the influenza incidence time series in US and Japan. Specifically, we use the aggregated and normalized counts of outpatients

exhibiting influenza-like symptoms released weekly by CDC[1]. For influenza in Japan, we use influenza-affected patient counts collected by NIID[2]. We forecast up to 4 weeks ahead over the period of 2004 to 2019 flu seasons using a similar setup as Flusight competitions Reich et al. (2019).

We also perform electricity forecasting on the ETT1 and ETT2 datasets using the train/test split mentioned previously. The last 10% of PEM-Bays dataset is used for traffic forecasting up to 1 hour ahead and the last 5 days of New York demand datasets for demand forecasting up to 120 minutes in the future. We also perform forecasting on the Nasdaq dataset for up to 5 days ahead and M3 time-series for 1 month ahead. We use 6 of the sensor datasets from Asuncion and Newman (2007) for time-series classification tasks. We use an 80-20 train-test split similar to Chowdhury et al. (2022).

**Baselines** We compared LPTM's performance in various time-series tasks against twenty two state-of-the-art general forecasting and domain-specific baselines. First, we compare against recent pre-trained foundational time-series models: (1) LLM-Time Gruver et al. (2024), (2) TimesFM Das et al. (2023b), (3) Lag-LLAMA Rasul et al. (2023), (4) Chronos Ansari et al. (2024) and (5) MOIRAI Woo et al. (2024). We skip models like Time-LLM Jin et al. (2024), MOMENT Goswami et al. (2024) and Autotimes Liu et al. (2024b) which cannot perform zero-shot forecasting across domains and are outperformed by the aforementioned recent models when fine-tuned. We compared with (6) Informer Zhou et al. (2021), (7) Autoformer Chen et al. (2021), (8) iTransformer Liu et al. (2023) and (9) PatchTST Nie et al. (2022), four state-of-the-art transformer-based forecasting models. We also compare against other recent model (10) MICN (Wang et al., 2022), (11) TiDE Das et al. (2023a) (12) TFT Lim et al. (2021) and (13) TimesNeT Wu et al. (2023). We also compare with (14) N-HITS Challu et al. (2023) which uses multi-scale interpolation and (15) AutoARIMA Hyndman and Khandakar (2008) a ARIMA based model that does automatic hyperparameter search. We also compare it against three other state-of-art self-supervised methods for time-series: (16) TS2Vec (Yue et al., 2022), (17) TS-TCC (Eldele et al., 2021) and (18) SimMTM Dong et al. (2024) uses masking as pre-trained task for time-series classification.

Finally, we compared against the best models for individual tasks for each domain. For influenza forecasting, we compared against previous state-of-art models (19) EpiFNP Kamarthi et al. (2021) and (20) ColaGNN Deng et al. (2020) respectively. We also compare against (21) STEP Shao et al. (2022) a SOTA model for demand forecasting, traffic prediction, and stock prediction benchmarks among the baselines by automatically modelling sparse relations between multiple features of the time-series. For classification tasks, we compare against (22) TARNet Chowdhury et al. (2022).

# 6 Results

Table 1: Average *zero-shot* forecast performance (measured as RMSE over 10 runs) of LPTM and *pre-trained* baselines. The best model is in **bold**.

| Model | Flu-US | Flu-japan | ETT1 | ETT2 | PEM-B | NY-B | NY-T | Nasdaq | M4 |
|---|---|---|---|---|---|---|---|---|---|
| LLM-Time | 1.38 | 1411 | 0.57 | 0.54 | 4.3 | 4.5 | 13.53 | 0.29 | 1.189 |
| TimesFM | 1.35 | 1259 | 0.61 | 0.59 | 3.9 | 3.9 | 13.11 | 0.29 | 1.211 |
| Lag-LLAMA | 1.52 | 1488 | 0.83 | 1.06 | 5.3 | 3.8 | 12.84 | 0.24 | 1.311 |
| Chronos | 1.29 | 1274 | 0.62 | 0.56 | 4.2 | 3.6 | 13.74 | 0.29 | 1.125 |
| MOIRAI | 1.39 | 1411 | 0.69 | 0.52 | 4.2 | 4.4 | 13.82 | 0.27 | 1.192 |
| TS2Vec | 1.94 | 1233.1 | 1.33 | 1.82 | 3.7 | 4.1 | 14.39 | 0.87 | 1.616 |
| TS-TCC | 2.17 | 1356.15 | 1.14 | 1.57 | 4.1 | 3.8 | 15.72 | 0.92 | 1.492 |
| SimMTM | 2.17 | 1356.15 | 1.14 | 1.57 | 4.1 | 3.8 | 15.72 | 0.92 | 1.492 |
| LPTM | **1.14** | **996** | **0.53** | **0.49** | **3.4** | **3.2** | **13.12** | **0.22** | **0.972** |

The code for implementation of LPTM and datasets are provided at anonymized link[3] and hyperparameters are discussed in the Appendix. LPTM consists of 10 layers for the transformer and overall has about 100M parameters which 2x to over 10x smaller than other pre-trained time-series models.

**Zero-shot forecasting** An important benefit of foundational models in language and vision is their ability to adapt to novel tasks without any fine-tuning in a zero-shot setting Brown et al. (2020). We

---

[1]https://gis.cdc.gov/grasp/fluview/fluportaldashboard.html
[2]https://www.niid.go.jp/niid/en/idwr-e.html
[3]https://github.com/AdityaLab/LPTM

Table 2: Average forecast performance (measured as RMSE over 10 runs) of LPTM and baselines over different domains. The best model is in **bold** and the second best is underlined.

| Model | Flu-US | Flu-Japan | ETT1 | ETT2 | PEM-B | NY-B | NY-T | Nasdaq | M4 | Rank |
|---|---|---|---|---|---|---|---|---|---|---|
| AutoARIMA | 2.14 | 1344 | 0.73 | 0.64 | 4.1 | 4.13 | 16.43 | 0.62 | 1.89 | 25.06 |
| Informer | 1.62 | 1139 | 0.57 | 0.71 | 3.1 | 2.89 | 12.33 | 0.83 | 1.055 | 15.89 |
| Autoformer | 1.41 | 1227 | 0.72 | 0.82 | 2.7 | 2.73 | 12.71 | 0.19 | 0.887 | 13.67 |
| PatchTST | 0.96 | 1113 | 0.52 | 0.63 | **2.5** | 2.64 | 11.95 | 0.15 | 0.877 | 7.5 |
| N-HITS | 1.42 | 1211 | 0.53 | 0.62 | 2.9 | 2.74 | 11.87 | 0.57 | 0.968 | 13.0 |
| TiDE | 1.21 | 1186 | **0.49** | 0.49 | 3.5 | 3.86 | 11.95 | 0.57 | 1.078 | 13.44 |
| MICN | 0.95 | 1145 | **0.49** | 0.57 | 3.6 | 2.61 | 11.56 | 0.13 | 0.931 | 7.77 |
| TimesNet | 1.04 | 1194 | 0.56 | 0.62 | 3.9 | 2.83 | 11.82 | 0.19 | 1.055 | 12.11 |
| TFT | 1.21 | 1876 | 0.52 | 0.51 | 4.6 | 2.95 | 12.55 | 0.24 | 1.18 | 16.11 |
| iTransformer | 1.14 | 1256 | 0.57 | 0.59 | 4.3 | 2.83 | 13.16 | 0.29 | 1.125 | 17.5 |
| LLM-Time | 1.21 | 1319 | 0.52 | 0.49 | 3.9 | 3.7 | 12.11 | 0.21 | 1.064 | 14.05 |
| TimesFM | 1.32 | 1214 | 0.58 | 0.49 | 3.7 | 2.8 | 12.19 | 0.22 | 1.07 | 13.44 |
| Lag-LLAMA | 1.46 | 1416 | 0.61 | 0.57 | 3.9 | 2.9 | 13.43 | 0.28 | 1.33 | 20.16 |
| Chronos | 1.21 | 1228 | 0.59 | 0.52 | 3.7 | 3.1 | 12.82 | 0.27 | 1.04 | 15.55 |
| MOIRAI | 1.31 | 1336 | 0.62 | 0.55 | 3.9 | 3.5 | 13.71 | 0.24 | 1.21 | 19.22 |
| STEP | 1.17 | 983 | 0.54 | 0.93 | 2.7 | 2.52 | **10.37** | **0.11** | 1.331 | 10.33 |
| EpiFNP | **0.52** | 872 | 0.81 | 1.25 | 4.1 | 2.98 | 12.11 | 0.28 | 1.281 | 16.77 |
| ColaGNN | 1.65 | 694 | 0.72 | 1.19 | 3.9 | 3.19 | 14.97 | 0.25 | 1.185 | 19.22 |
| TS2Vec | 1.85 | 905.9 | 0.99 | 1.74 | 3.5 | 3.11 | 13.48 | 0.94 | 1.344 | 21.94 |
| SimMTM | 1.31 | 1289 | 0.61 | 0.55 | 3.4 | 3.1 | 12.79 | 0.28 | 1.284 | 17.94 |
| TS-TCC | 1.94 | 1134.6 | 0.75 | 1.29 | 3.3 | 2.97 | 15.55 | 0.76 | 1.274 | 21 |
| LPTM | 0.79 | 704 | **0.49** | **0.46** | **2.5** | 2.37 | 11.84 | 0.17 | **0.872** | 2.55 |
| LPTM-NoSegment | 0.93 | 766 | 0.57 | 0.55 | 3.2 | 3.17 | 14.96 | 0.27 | 1.146 | 13.72 |
| LPTM-NoPreTrain | 0.96 | 827 | 0.46 | 0.57 | 3.7 | 2.66 | 12.43 | 0.25 | 1.271 | 11.66 |
| LPTM-NoLinProb | 0.92 | 885 | 0.43 | 0.53 | 3.1 | 2.49 | 12.17 | 0.19 | 1.032 | 6.55 |
| LPTM-OnlyRandMask | 0.87 | 895 | 0.51 | 0.52 | 2.8 | 2.42 | 12.36 | 0.21 | 1.076 | 8.0 |
| LPTM-OnlyLastMask | 0.79 | 773 | 0.44 | 0.48 | 2.7 | 2.31 | 12.04 | 0.19 | 0.916 | **3.77** |

evaluate the zero-shot performance of LPTM and other pre-trained baselines. Similar to Gruver et al. (2024) we use the last 20% of the datasets for zero-shot evaluation. We do not fine-tune the models but only input the normalized time-series for new tasks directly during inference. The results are summarized in Table 1. LPTM outperforms all the baselines significantly. Moreover, the baselines such as TS2Vec, TS-TCC and SimMTM which are not designed to handle datasets from multiple domains perform much worse than other pre-trained methods. This shows the importance of adaptive segmentation-based tokenization of LPTM to better generalize across multiple domains.

**Fine-tuned forecasting**  We also train the models with training data to evaluate fine-tuned performance. The forecasting performance are shown in Table 2. We evaluated LPTM against seventeen other forecasting baselines. LPTM is either the first or a close second best-performing model in all the benchmarks. LPTM generally outperforms general time-series forecasting methods as well as 2recent pre-trained models in all benchmarks despite its much lower parameter count (2-10x lower). Further, it is competitive or superior in performance to domain-specific methods designed specifically for the given domains (such as EpiFNP and STEP). LPTM beats the previous state-of-art domain-specific baselines in five of the benchmarks and comes second in four more. Finally, LPTM improves upon the state-of-art on electricity forecasting, traffic forecasting, and M3 datasets.

**Time-series classification**  Unlike other autoregressive foundational models designed for forecasting, LPTM can be used for classification due to its encoder-style architecture. We add a single classification layer over pooled output embeddings $\{o^{(i)}\}_{i=1}^{R}$ to predict the class logits. We evaluate LPTM and baselines on the classification of 35 sensor and behavioral datasets from UCI classification repositiry(Asuncion and Newman, 2007). We report the accuracy scores in Table 3. We observe that LPTM has highest mean rank and largest number of times it outperforms all baselines.

**Data efficiency**  A significant advantage of leveraging pre-trained models is that we do not require large datasets for fine-tuning to a specific task. We evaluate the efficacy of LPTM to train with a fraction of training data. For each time-series analysis task, we fine-tune the model using only $k\%$ of training data for different values of $k$. We use the first $k\%$ of the timestamps' values. We do not choose a random sample to prevent data mixing from the rejected portion of training data. We also performed the similar experiment on the best baseline for each task and compare data efficiency of baseline with LPTM. The comparison plots are shown in Figure 2. With lesser data, the performance

Table 3: Average classification performance (measured as accuracy score over 10 runs) of LPTM and baselines over different domains. The best model is in **bold** and the second best is underlined. The best model is statistically significant over the baselines ($p \leq 0.05$) when it beats the previous state-of-art.

| | Informer | Autoformer | TimesNet | TARNet | TS2Vec | TS-TCC | TST | SimMTM | CRT | LPTM |
|---|---|---|---|---|---|---|---|---|---|---|
| BasicMotions | 0.95 | 0.93 | 0.92 | **1.00** | 0.99 | **1.00** | 0.92 | 0.86 | 0.88 | **1.00** |
| FaceDetection | 0.51 | 0.49 | 0.59 | 0.63 | 0.51 | 0.54 | 0.62 | 0.73 | 0.78 | **0.79** |
| FingerMovements | 0.58 | 0.54 | 0.58 | 0.62 | 0.46 | 0.47 | 0.59 | 0.68 | 0.72 | **0.78** |
| PEMS-SF | 0.67 | 0.71 | 0.84 | **0.94** | 0.75 | 0.73 | 0.93 | 0.86 | 0.89 | 0.93 |
| RacketSports | 0.83 | 0.86 | 0.91 | **0.98** | 0.77 | 0.85 | 0.79 | 0.84 | 0.87 | 0.93 |
| EigenWorms | 0.49 | 0.62 | 0.73 | 0.89 | 0.84 | 0.77 | 0.72 | 0.82 | 0.79 | **0.94** |
| ArticularyWordRecognition | 0.83 | 0.82 | 0.79 | 0.97 | 0.89 | 0.97 | 0.92 | 0.92 | 0.88 | **0.98** |
| AtrialFibrillation | 0.57 | 0.55 | 0.68 | **1.00** | 0.44 | 0.37 | 0.72 | 0.85 | 0.89 | 0.93 |
| CharacterTrajectories | 0.57 | 0.55 | 0.68 | 0.97 | 0.98 | 0.96 | **0.99** | 0.97 | 0.93 | 0.98 |
| Cricket | 0.94 | 0.87 | 0.88 | **1.00** | 0.98 | 0.97 | 0.84 | 0.96 | 0.94 | 0.99 |
| DuckGeese | 0.54 | 0.44 | 0.56 | 0.75 | 0.39 | 0.57 | 0.74 | 0.58 | 0.55 | **0.79** |
| Epilepsy | 0.58 | 0.57 | 0.61 | **1.00** | 1.00 | 0.98 | 0.94 | 0.78 | 0.55 | 0.97 |
| ERing | 0.23 | 0.29 | 0.46 | 0.92 | 0.89 | 0.78 | 0.86 | 0.64 | 0.75 | **0.97** |
| EthanolConcentration | 0.14 | 0.27 | 0.34 | 0.32 | 0.45 | 0.37 | 0.46 | 0.34 | 0.21 | **0.53** |
| HandMovementDirection | 0.14 | 0.27 | 0.34 | 0.32 | 0.45 | 0.37 | 0.46 | 0.34 | 0.21 | **0.53** |
| Handwriting | 0.16 | 0.18 | 0.29 | 0.24 | 0.52 | 0.55 | 0.37 | **0.64** | 0.52 | 0.51 |
| Heartbeat | 0.53 | 0.66 | 0.61 | **0.78** | 0.71 | 0.69 | 0.74 | 0.74 | 0.62 | 0.74 |
| InsectWingbeat | 0.13 | 0.16 | 0.14 | 0.14 | 0.18 | 0.22 | 0.69 | 0.62 | 0.18 | **0.72** |
| JapaneseVowels | 0.87 | 0.96 | 0.94 | **0.99** | 0.97 | 0.98 | **0.99** | 0.94 | **0.99** | 0.98 |
| Libras | 0.72 | 0.64 | 0.75 | **1.00** | 0.85 | 0.92 | 0.91 | 0.82 | 0.88 | 0.95 |
| LSST | 0.36 | 0.44 | 0.32 | 0.97 | 0.54 | 0.62 | 0.65 | 0.53 | 0.48 | **0.98** |
| MotorImagery | 0.51 | 0.52 | 0.50 | **0.64** | 0.62 | 0.61 | 0.62 | 0.46 | 0.58 | 0.57 |
| NATOPS | 0.75 | 0.69 | 0.84 | 0.92 | 0.91 | 0.92 | **0.94** | 0.85 | 0.88 | **0.94** |
| PenDigits | 0.84 | 0.86 | 0.81 | 0.97 | **0.98** | 0.95 | 0.97 | 0.94 | 0.88 | 0.92 |
| Phoneme | 0.11 | 0.13 | 0.15 | 0.17 | 0.28 | 0.226 | 0.29 | 0.26 | 0.18 | **0.32** |
| SelfRegulation | 0.65 | 0.76 | 0.57 | 0.81 | 0.77 | 0.84 | 0.89 | **0.96** | 0.73 | 0.92 |
| SpokenArabicDigits | 0.92 | 0.96 | 0.92 | 0.98 | 0.94 | 0.97 | 0.99 | 0.96 | 0.94 | **1.00** |
| StandWalkJump | 0.21 | 0.10 | 0.34 | 0.53 | 0.28 | 0.31 | **0.61** | 0.52 | 0.11 | 0.58 |
| UWaveGesture | 0.79 | 0.86 | 0.82 | 0.87 | 0.91 | 0.92 | 0.86 | 0.74 | 0.82 | **0.94** |
| PAMAP2 | 0.73 | 0.87 | 0.84 | **0.97** | 0.93 | 0.94 | 0.96 | 0.93 | 0.88 | **0.97** |
| OpportunityGestures | 0.73 | 0.66 | 0.68 | 0.83 | 0.92 | 0.74 | 0.74 | 0.62 | 0.72 | **0.92** |
| OpportunityLocomotion | 0.84 | 0.78 | 0.85 | **0.91** | 0.75 | 0.82 | 0.84 | 0.87 | 0.89 | 0.89 |
| SelfRegulationSCP2 | 0.562 | 0.598 | 0.612 | 0.622 | 0.593 | 0.575 | 0.604 | 0.614 | 0.625 | 0.691 |
| Occupancy | 0.774 | 0.739 | 0.814 | 0.833 | 0.876 | 0.865 | 0.881 | 0.826 | 0.814 | 0.836 |
| MosquitoSound | 0.439 | 0.493 | 0.551 | 0.632 | 0.649 | 0.662 | 0.691 | 0.703 | 0.624 | 0.715 |

Figure 2: Performance of LPTM and best baseline with varying fractions of training data. In most cases LPTM significantly outperforms baselines with lower amount of data.

of the baseline is much worse whereas LPTM typically requires much less data to provide similar performance to when we have access to the full dataset. This shows the importance of pre-training to quickly ramp up the performance of the model with much less data, a problem we encounter is many real-world settings such as when we need to deploy a forecasting model on novel applications such as a new pandemic with sparse data availability.

**Training efficiency** An important advantage of pre-trained models is that they require much less training time and resources to fine-tune to a downstream task compared to time required for pre-training or even training from scratch. We compare the fine-tuning time of LPTM with baselines on benchmarks from different domains. We also measure the average time required by LPTM to reach the performance of the best baseline in cases where we eventually outperform them. The training times are summarized in Appendix Table 4. We observe that the time taken by LPTM to reach the performance of best best-performing baseline (LPTM-TB) is significantly smaller than the time taken by any other baselines. Even when LPTM doesn't outperform the best baseline, it typically converges much faster.

**Ablation and Sensitivity Studies**   We study the impact of our adaptive segmentation as well as pre-training and linear probing via the ablation models LPTM-NoSegment, LPTM-NoPreTrain and LPTM-NoLinProb. We also investigate the individual impact of both SSL task via the ablation models LPTM-OnlyRandMask and LPTM-OnlyLastMask. The performance of the ablation variants are also shown in Tables 2. We observe that the ablation variants' performances are significantly worse than the variants, underperforming some of the baselines. The worst performing variant is usually LPTM-NoSegment, showing the importance of deriving good time-series segments to improve representation learning of time-series for each dataset. We also examined the sensitivity of hyperparameter $\gamma$ for SSL tasks and found the optimal value at 0.4 for LASTMASK and 0.2 for RANDMASK. The sensitivity analysis plots are in Appendix Fig. 5.

**Segments generated by LPTM**   We also visualized the segments in Fig. 3. We observe that the segment sizes are smaller in regions of high variance or important parts of the time-series such as peak of the epidemic whereas simpler trends have longer segments which matches our intuition.

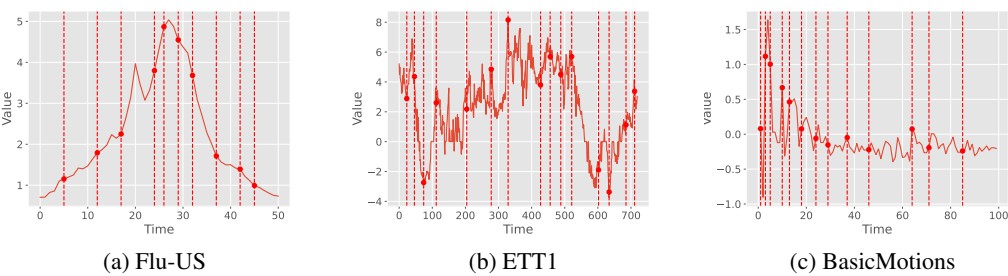

|       (a) Flu-US       |       (b) ETT1       |       (c) BasicMotions       |

Figure 3: Segmentation learned by LPTM

# 7   Conclusion

We make a significant contribution towards general pre-trained models for time-series analysis tasks replicating the success of large pre-trained models in language and vision domains. We introduce LPTM, a general pre-trained model that provides state-of-art performance on a wide range of forecasting and classification tasks from varied domains and applications. LPTM provides similar to or better performance to state-of-art domain-specific models in applications such as epidemiology, energy, traffic, and economics and significantly beats recent time-series foundational models. We also observe that LPTM required significantly lesser pre-training and training data to reach optimal performance compared to other baselines in most benchmarks.

Our work mainly focuses on the important challenge of providing semantically meaningful inputs to the model that caters to learning time-series segmentation strategies specific to each domain. This is crucial when pre-training on diverse datasets, a key challenge for time-series data. The underlying model architecture is a straightforward transformer encoder that uses well-known masking techniques for self-supervised pre-training. Therefore, our method can be extended to leverage novel time-series model architectures and SSL methods. Extending our methods to provide calibrated forecasts that provide reliable uncertainty measures is also another important direction of research. We can also extend it to leverage multimodal datasets like text that provide important contextual information about the dataset Liu et al. (2024c).

Since our model can be applied to any generic time-series analysis tasks including those in critical domains such as public health, medicine, economics, etc., important steps need to be taken to address potential misuse of the our methods such as testing for fairness, data quality issues, ethical implications of predictions, etc.

## Acknowledgements

This paper was supported in part bythe NSF (Expeditions CCF-1918770, CAREER IIS-2028586, Medium IIS-1955883, Medium IIS-2403240, Medium IIS-2106961, PIPP CCF-2200269), CDC MInD program, Meta faculty gifts, and funds/computing resources from Georgia Tech.

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

# Appendix for Large Pre-trained time series models for cross-domain Time series analysis tasks

## A  Hyperparameters

The model is run on Intel Xeon CPU with 64 cores and 128 GB RAM. We use a single A100 GPU with 80GB memory. For GRU we use a single hidden layer of 50 hidden units. Dimension of $\mathbf{v}$ is also 50. The transformer architecture consists of 10 layers with 8 attention heads each. For forecasting tasks, we train a separate decoder module with 4 more layers during fine-tuning whereas for classification we aggregate the embeddings $\{e_i\}_{i=1}^{R}$ of the last transformer layer and feed them into a single linear layer that provides logits for all classes. The SSL pre-training was done till convergence via early stopping with patience of 1000 epochs. We observed that LPTM takes 5000-8000 epochs to finish pre-training which takes around 3-4 hours. (Note that pre-training is a one-time step and downstream fine-tuning takes much less time and epochs). For both pre-training and fine-tuning, we used the Adam optimizer with a learning rate of 0.001. The hyperparameters are tuned sparingly for both LPTM and baselines from their default settings. For RANDMASK, we found the optimal $\gamma = 0.4$, and for LASTMASK $\gamma = 0.2$ was optimal.

## B  Data efficiency

Table 4: Average training time (minutes) till convergence for LPTM and neural baselines. LPTM-TB shows the time taken by LPTM to reach performance of top baseline (in benchmarks where LPTM outperforms it). Since some baselines are specific to forecasting or classification and we do not beat the state-of-art in few benchmarks we designate these cells in the table as NA.

| Model | Flu-US | ETT2 | PEM-Bays | NY-Bike | Nasdaq | M3 | BasicMotions | EigenWorms |
|---|---|---|---|---|---|---|---|---|
| Informer | 27.3 | 25.5 | 45.1 | 49.7 | 27.1 | 49.6 | 17.5 | 14.3 |
| Autoformer | 19.5 | 29.3 | 49.5 | 55.2 | 18.5 | 45.1 | 11.9 | 19.7 |
| N-HITS | 15.4 | 22.5 | 36.1 | 49.3 | 26.4 | 47.5 | NA | NA |
| PatchTST | 12.9 | 29.5 | 36.4 | 45.7 | 18.2 | 49.4 | NA | NAg |
| MICN | 17.6 | 15.7 | 39.7 | 41.1 | 19.2 | 33.9 | NA | NA |
| TimesNet | 15.4 | 19.7 | 37.4 | 46.3 | 24.1 | 36.5 | 9.4 | 11.5 |
| STEP | 25.4 | 34.1 | 52.7 | 74.3 | 29.7 | 52.8 | NA | NA |
| EpiFNP | 22.5 | 39.5 | 41.1 | 39.1 | 21.6 | 97.6 | NA | NA |
| ColaGNN | 34.7 | 33.6 | 53.1 | 47.6 | 32.1 | 72.2 | NA | NA |
| TARNet | NA | NA | NA | NA | NA | NA | 13.7 | 9.4 |
| TS2Vec | 29.3 | 28.2 | 41.9 | 41.9 | 29.8 | 67.4 | 9.3 | 13.2 |
| TS-TCC | 21.7 | 23.7 | 46.3 | 44.3 | 25.3 | 55.8 | 12.7 | 11.1 |
| LPTM | **12.2** | 19.3 | 41.9 | **37.5** | **17.3** | **31.2** | **6.1** | 12.7 |
| LPTM-TB | NA | **12.5** | **29.6** | **32.9** | NA | **23.7** | **6.1** | **8.1** |

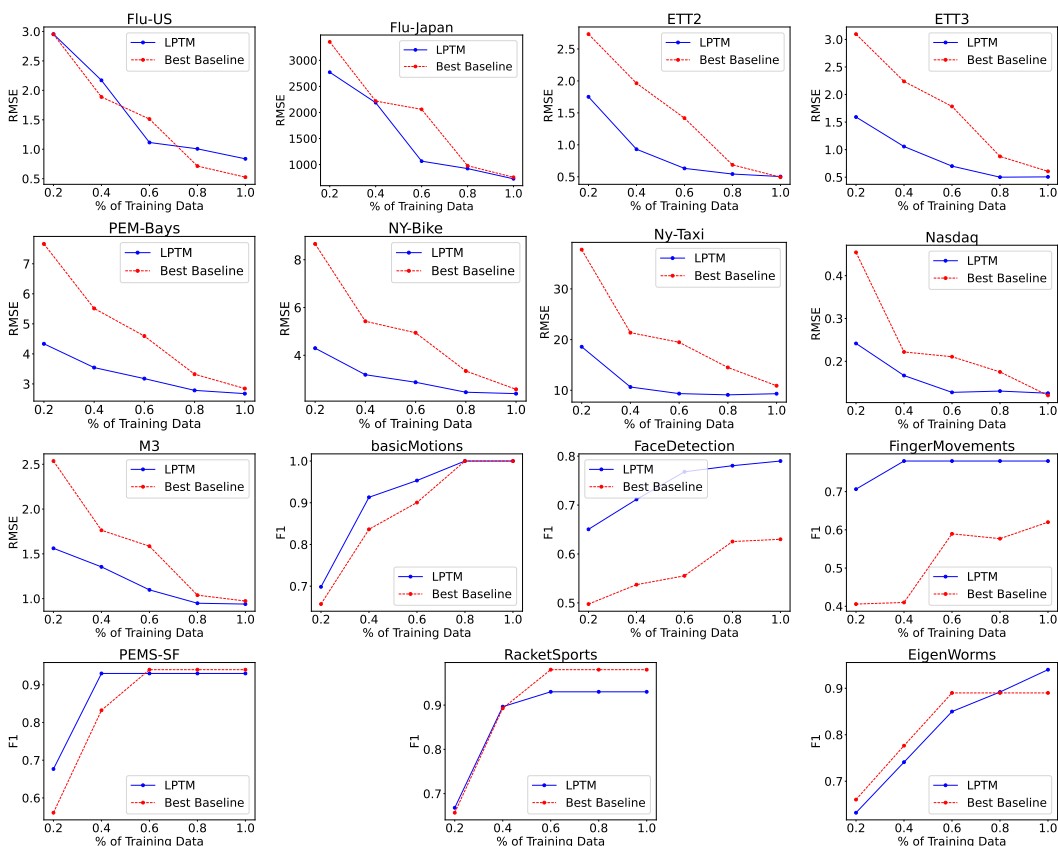

Figure 4: Performance of LPTM and best baseline with varying fractions of training data. In most cases LPTM significantly outperforms baselines with lower amount of data.

# C   Effect of SSL hyperparameter $\gamma$

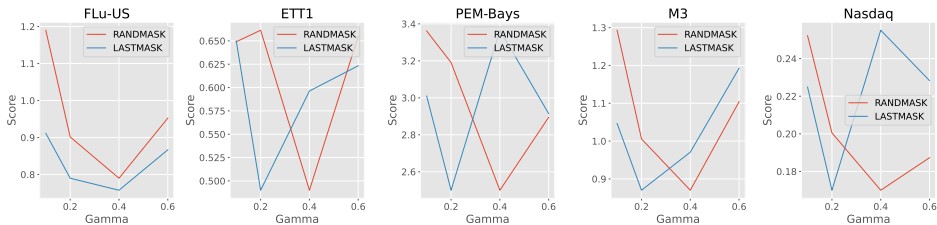

Figure 5: Effect of $\gamma$ on performance(RMSE) for different benchmarks

Table 5: Std. dev across 10 runs

| Model | Flu-US | Flu-Japan | ETT1 | ETT2 | PEM-Bays | NY-Bike | NY-Taxi | Nasdaq | M4 |
|---|---|---|---|---|---|---|---|---|---|
| AutoARIMA | 0.043180 | 23.434077 | 0.014220 | 0.035201 | 0.056066 | 0.044572 | 0.151146 | 0.025181 | 0.024060 |
| Informer | 0.044072 | 23.918021 | 0.014514 | 0.035927 | 0.057224 | 0.045492 | 0.154268 | 0.025701 | 0.024557 |
| Autoformer | 0.042310 | 22.961906 | 0.013934 | 0.034491 | 0.054936 | 0.043674 | 0.148101 | 0.024673 | 0.023575 |
| PatchTST | 0.043320 | 23.509948 | 0.014266 | 0.035314 | 0.056248 | 0.044716 | 0.151636 | 0.025262 | 0.024138 |
| N-HITS | 0.048238 | 26.178970 | 0.015886 | 0.039324 | 0.062633 | 0.049793 | 0.168850 | 0.028130 | 0.026878 |
| MICN | 0.030538 | 16.573161 | 0.010057 | 0.024895 | 0.039651 | 0.031522 | 0.106894 | 0.017808 | 0.017016 |
| TimesNet | 0.021264 | 11.539913 | 0.007003 | 0.017334 | 0.027609 | 0.021949 | 0.074431 | 0.012400 | 0.011848 |
| LLM-Time | 0.040279 | 21.859627 | 0.013265 | 0.032836 | 0.052299 | 0.041577 | 0.140991 | 0.023489 | 0.022443 |
| TimesFM | 0.034938 | 18.961321 | 0.011506 | 0.028482 | 0.045365 | 0.036065 | 0.122298 | 0.020375 | 0.019468 |
| Lag-LLAMA | 0.028790 | 15.624783 | 0.009481 | 0.023470 | 0.037382 | 0.029719 | 0.100777 | 0.016789 | 0.016042 |
| Chronos | 0.040037 | 21.728648 | 0.013185 | 0.032639 | 0.051986 | 0.041328 | 0.140146 | 0.023348 | 0.022309 |
| STEP | 0.028283 | 15.349319 | 0.009314 | 0.023056 | 0.036723 | 0.029195 | 0.099001 | 0.016493 | 0.015759 |
| EpiFNP | 0.026098 | 14.163743 | 0.008595 | 0.021275 | 0.033887 | 0.026940 | 0.091354 | 0.015219 | 0.014542 |
| ColaGNN | 0.038447 | 20.865267 | 0.012661 | 0.031342 | 0.049920 | 0.039686 | 0.134578 | 0.022421 | 0.021422 |
| TS2Vec | 0.032028 | 17.381828 | 0.010548 | 0.026109 | 0.041586 | 0.033061 | 0.112110 | 0.018677 | 0.017846 |
| SimMTM | 0.053732 | 29.160611 | 0.017695 | 0.043802 | 0.069767 | 0.055464 | 0.188081 | 0.031334 | 0.029939 |
| TS-TCC | 0.044399 | 24.095686 | 0.014622 | 0.036194 | 0.057649 | 0.045830 | 0.155414 | 0.025892 | 0.024739 |
| LPTM | 0.048677 | 26.417645 | 0.016031 | 0.039682 | 0.063204 | 0.050247 | 0.170390 | 0.028387 | 0.027123 |
| LPTM-NoSegment | 0.045970 | 24.948198 | 0.015139 | 0.037475 | 0.059689 | 0.047452 | 0.160912 | 0.026808 | 0.025614 |
| LPTM-NoPreTrain | 0.058905 | 31.968143 | 0.019399 | 0.048020 | 0.076484 | 0.060804 | 0.206190 | 0.034351 | 0.032822 |
| LPTM-NoLinProb | 0.025968 | 14.092985 | 0.008552 | 0.021169 | 0.033718 | 0.026805 | 0.090898 | 0.015143 | 0.014469 |
| LPTM-OnlyRandMask | 0.039517 | 21.446157 | 0.013014 | 0.032214 | 0.051310 | 0.040791 | 0.138324 | 0.023045 | 0.022019 |
| LPTM-OnlyLastMask | 0.058067 | 31.513665 | 0.019123 | 0.047337 | 0.075397 | 0.059940 | 0.203258 | 0.033863 | 0.032355 |

| Model | Score |
|---|---|
| AutoARIMA | 21.388889 |
| Informer | 14.055556 |
| Autoformer | 12.000000 |
| PatchTST | 6.833333 |
| N-HITS | 11.388889 |
| MICN | 7.277778 |
| TimesNet | 11.111111 |
| LLM-Time | 12.777778 |
| TimesFM | 12.277778 |
| Lag-LLAMA | 17.444444 |
| Chronos | 13.888889 |
| STEP | 9.111111 |
| EpiFNP | 14.666667 |
| ColaGNN | 16.500000 |
| TS2Vec | 19.111111 |
| SimMTM | 15.611111 |
| TS-TCC | 18.111111 |
| LPTM | 2.500000 |
| LPTM-NoSegment | 12.055556 |
| LPTM-NoPreTrain | 10.444444 |
| LPTM-NoLinProb | 6.222222 |
| LPTM-OnlyRandMask | 7.555556 |
| LPTM-OnlyLastMask | 3.666667 |

Table 6: Mean rank of models in Table 2

