# OpenReview forum: "Large Pre-trained time series models for cross-domain Time series analysis tasks"
_NeurIPS.cc/2024/Conference — NeurIPS 2024 poster_

### Official Review · Reviewer_fxq9 · 2024-07-08

**Soundness:** 3
**Presentation:** 2
**Contribution:** 3
**Rating:** 5
**Confidence:** 4

**Summary:**

Training large time series (TS) models is often limited by the scarce data available for a specific application. Existing pretraining methods use a simplistic tokenization scheme where the TS is cut up into equally sized parts, independent of its content. The newly proposed method *Large Pre-trained Time-series Models*, therefore, adaptively segments the input time series into (potentially) overlapping tokens depending on the TS itself. It shows very good forecasting performance in zero-shot and finetuning settings. It can also be used for classification.

**Strengths:**

- The relevance of the problem and motivation for adaptive segmentation is convincing.
- The method of adaptive segmentation is an interesting solution to the issue.
- LPTM is compared against a plethora of appropriate and challenging baselines.
- It shows promising empirical results.

**Weaknesses:**

- I am under the impression that this paper may have found a strong method yet does not sufficiently investigate *why* it works. The interplay between learning the scoring function and training the encoder is not very clear. See below.
- A lot of experimental claims are not adequately substantiated. It is claimed in question 6 of the checklist that error bars are provided and that statistical significance tests are performed, yet I did not find them. See below.
- The overall presentation (language and formatting) should be improved.
- The provided implementation is not accessible. (Error: "The repository is expired") In the current state, results are not reproducible since key hyperparameters are missing. The authors claim in question 6 of the checklist that they state how hyperparameters were chose, yet I could not find it in the paper.

**Questions:**

1. The problem statement (l. 96-98) only considers in-domain applications. Is it also interesting to consider domain generalization settings?
2. The term "Large Pre-trained Time-series Models (LPTM)" (l. 48-51) is extremely generic and better describes the emerging field than a single method. Would it make sense to focus more on the "Adaptive Segmentation" contribution in the title and exposition?
3. Why did you choose Eq. (2) to be that way? Are all parameters learned?
4. The scaling of the method is not discussed sufficiently; could you comment on that? There are potentially a lot of tokens -- in the worst case, there are as many as input tokens. Even more, the number of evaluations of $s(i,j)$ scales quadratically in the input length. While the heuristic for selecting a good set of segments is defined well, a discussion of why it is sensible is missing. See also question 7.2.
5. Why did you use masked reconstruction as the two pretraining tasks? Could you explain why this is more desirable than alternatives, like contrastive methods?
6. The interplay of training the segmentation strategy and the encoder simultaneously requires a more nuanced discussion. To what extent can this cause something like a "mode collapse" (as in Generative Adversarial Networks), where the $s(i,j)$ always chooses the same segments since they were found to be beneficial at some point and stops "exploring" others? This should (1) be discussed in more depth and (2) may be a significant limitation of the method.
7. This leads to the experimental evaluation.
	1. Following (6), can you provide insights into the relative convergence speeds of the scorer and the encoder/decoder? How good is the score function $s$ at predicting the final encoder loss?
	2. How many segments are there typically? A histogram would help judge the typical number of tokens resulting from the adaptive segmentation.
	3. L. 281: "2x to over 10x smaller than other pre-trained time-series models" -> Where do you provide that comparison of model sizes?
	4. Section 5 only discusses (almost) exclusively the forecasting setup. Why were the specific classification datasets chosen? Section 6 mentions 35 datasets (UCI has more time series classification datasets), but Table 4 only contains 32. Were three datasets removed at some point?
	5. L. 305: "We observe that LPTM has highest mean rank" -> Please provide that rank in (or alongside) Table 4.
	6. The caption of Table 4 talks about statistical significance. What are the test's results? Which significance test did you even perform? How were results aggregated (if it was the arithmetic mean, what is their std. dev.)?

Minor comments:
- The paper would benefit from a language pass.
- Improvements to formatting: References are formatted confusingly, e.g., in lines 22, 23f, 36f, etc. This problem occurs throughout the paper. Table 4 in the Appendix is too wide. REVIN -> RevIN. Formatting of LASTMASK, etc., is inconsistent (l. 163 vs. l. 335), ...
- The example in line 40ff could be given more bluntly and convincingly by discussing the same event in different sample rates.
- L. 91, what is R? The same as in l. 152?
- L. 89, isn't $\mathcal{D}_\text{pre}$ the union over the individual datasets?
- The abbreviation SSL has never been introduced.
- Red and green are difficult to distinguish for people with common color deficiencies. Therefore, Figure 1 could be improved with a different color palette.
- Table 3 should mention that it is about finetuning. Are the baselines trained from scratch or merely finetuned as well?
- The authors should check whether they intentionally want to cite many preprints (e.g., from Arxiv) or their published variants.
- The capitalization of the title is inconsistent.

**Limitations:**

Depending on the answers to the questions above, possible limitations mentioned could be made more transparent. For example, the insights into the newly induced biases are currently limited.

The answer to section 2. in the checklist is neither sufficient nor truthful. For example, "multivariate" is never mentioned in Section 7.

The discussion of the societal impact could also consider a possible data leakage from one private application (e.g., in the medical domain) to another one. Even a rather mundane problem, like a feasible membership inference attack, could be problematic in privacy-sensitive scenarios.

---

> ### Author Rebuttal · Authors · 2024-08-07
>
> We thank the reviewer for their comments. We will address them as follows:
>
> **Domain-generalization settings**
>
> We wish to emphasize that we generalize to a wide range of domains but we require to learn segmentation module for each of the domains.
> Generalizing to unseen domains is an important research question to tackle for general pre-trained model
>
> **Title**
>
> We chose LPTM as the title since our innovation using adaptive segmentation for cross-domain pre-training enables state-of-art performance across multiple datasets with heterogenous patterns. We are happy to add to adaptive segmentation to the title.
>
> Moreover, other pre-trained baselines also have generic names such as TimesFM (Time-series foundational model).
>
> **Eq 2**
>
> The intuition for Eq 2 is that since the GRU captures the information at timesteps i and j in context of entire time-series, the segment score function uses this information encoded in z^(i) and z^(j) to derive the segment score. The exact form of the equation was chosen empirically and is similar in form to Bahdanau attention function. All the parameters (W_1, W_2 and b) are learnable.
>
> **Scaling and number of tokens**
>
> Actually, we found that adaptive segmentation doesn’t provide any significant contribution to overall overhead of the model since the transformer backbone consumes the bulk of the compute time. While the entire adaptive segmentation module’s runtime is quadratic in length of the time-series, Empirically we found that the average size of the segments across datasets is around 5.2 with some domains such as ETT having larger average lengths (13.4) and some domains such as behavioral datasets with lower average lengths (3.1).
>
> Regarding the heuristic, solving for the optimal set of segments that provide the maximum score as well as cover the entire length of the input is a non-trivial optimization problem whose complexity could be upto exponential in number of time-steps. Therefore, we use a simple heuristic of eliminating the lowest valued segments till we cannot cover the entire time-series. (l 12-14).
>
>
> **Choice of SSL tasks**
>
> Randon masking and lask token predictions are popular SSL pre-training methods that are widely adopted in LLMs and some time-series methods like PatchTST. We also point out that using these simple tasks already provides state-of-art results.
> While we can use more sophisticated methods, the focus of our work is on using adaptive segmentation to enable multi-domain pre-trained time-series model. Indeed LPTM is very flexible to adapt other SSL loss functions to improve performance could be a interesting extension of our work.
>
> **Stability of training and mode collapse**
>
> We did not observe instability or mode collapse during pre-training. We found that the training of the model was stable across different domains and datasets. This was observed across multiple runs of the model as well. The frequency of updating the segmentation loss relative to SSL loss is key in ensuring the stability of training.
>
> The overall objective of self-supervised pre-training involves a discrete objective of choosing an optimal segmentation strategy along with the prediction of masked segments. Similar to other works in graph learning[1] and reinforcement learning on discrete action space [2] which involve learning over discrete structures that cannot be trivially integrated we use a bilevel optimization approach, and periodically optimize the segmentation by learning on SSL tasks.
>
> Improving the efficiency and stability of this approach and more tightly coupling the two objectives could be an important research direction to improve model performance.
>
> **Convergence of scorer and encoder**
>
> It is not straightforward to compare the SSL loss with the loss for score function since they indicate different objectives. We found that the SSL loss may slightly perturb when there is an update to score function loss but generally follows a relatively smooth downward trend to convergence. The scoring function loss, while trending lower over time, doesn't converge to zero and hence is not a predictor of SSL loss, Eq 5 only forces the scoring function to be trained to update in the direction of decreasing SSL loss.
>
> **Model Sizes**
>
> LPTM model size is around 100M which is 2x smaller than than TimesFM, the smallest of the pre-trained baselines and is around 10x smaller than Chronos which has around 1B parameters.
>
> **Classification tasks**
>
> While we focused on forecasting for most of the experiments similar to other general pre-trained time-series models, we also showcased the flexibility of LPTM to perform classification due to our architectural choice.
> Table 4 contains only 32 datasets and this is a typo. We added the other 3 datasets in the additional pdf.
>
> **Mean rank of Models**
>
> We post the mean ranks in the additional PDF. LPTM has the best average rank of 2.5 and is significantly better than other models.
>
> **Significance test**
>
> We used the standard t-test to determine statistical significance. We have added the standard deviations across 10 runs in the additional pdf.
>
> **Regarding Limitations**
>
> We thank the reviewer for the suggestions. We will add the discussion on multivariate datasets and issues regarding data privacy to discussions.
>
> **Code link**
>
> Sorry. We didn’t realize the link was expired during the review process. The code can be found in the LPTM.zip file at https://osf.io/vy3sc/?view_only=c10508d340ba468287c984496cf57be1

---

> > ### Comment · Reviewer_fxq9 · 2024-08-11
> >
> > Thank you for your insightful responses. I have no further major questions. I hope the above discussion and improvements to the language and general presentation make it into the camera-ready version.
> >
> > **I raised my score from 3 to 5.**
> >
> > As a side note, providing Critical Difference diagrams for the classification results would improve the rigor of the analysis.

---

> > > ### Author Response · Authors · 2024-08-11
> > >
> > > We thank the reviewer for acknowledging our response and are glad to have addressed their concerns. We are grateful for increasing their scores.

---

### Official Review · Reviewer_RuJ6 · 2024-07-12

**Soundness:** 3
**Presentation:** 2
**Contribution:** 3
**Rating:** 7
**Confidence:** 4

**Summary:**

The paper introduces a new approach for creating pre-trained models for time-series data, similar to those used in language and vision tasks. The authors propose a model called Large Pre-trained Time-series Models (LPTM), which includes an innovative adaptive segmentation module to handle diverse time-series data from multiple domains.

Key contributions include:

- Developing a framework for pre-training time-series models on multi-domain datasets, using a novel adaptive segmentation module to tokenize inputs effectively. This is achieved via a self-supervised learning objective.
- Demonstrating that LPTM performs as well or better than state-of-the-art domain-specific models when fine-tuned for various time-series tasks, such as forecasting and classification, with less training data and compute time.
- Proving that LPTM achieves superior results in both zero-shot and fine-tuned settings across diverse domains like epidemiology, energy, and economics, requiring up to 40% less data and 50% less training time compared to existing methods.

**Strengths:**

The paper has the following strengths:

- Well-written, clear, easy to follow. Algorithm is a nice plus.
- Baseline choice reasonable: most recent methods are considered.
- Experimental results good, when considered on the set of datasets chosen (more points on that in the weaknesses section).

**Weaknesses:**

- It's a bit hard to get a good feel for the relative advantage of the proposed method. In table 2, the approach is clearly better, but we are left to infer that from that fact that it is commonly second or first in the rankings. Could the authors maybe add some for of aggregate metric, e.g. the average rank across datasets of a given method?
- Despite mentioning code is available, the link does not work (subscript 3 on page 7, time of access 2024-07-12, and previously): "The repository is expired".
- For a paper dealing in large part with forecasting, I was surprised by the absence of almost all of the classical long-term forecasting datasets used by other papers: traffic, electricity, weather, illness... Given that these are by far the most heavily studied ones in the literature, including them (as proposed in the questions section). While I don't find it a critical (but still important) concern, I strongly advise the authors to consider adding them as it will help avoid concerns other readers might have about cherry-picking of results.

**Questions:**

- Can the authors add error bars (in the appendix possibly) for their experiments? They mention that they already run 10 experiments per setup so these should be readily available and would give a good idea of the robustness of the findings.
- Can the authors ensure that code to reproduce their experiments is available as stated?
- Can the authors run the same experiments on the "standard" long-term forecasting datasets, as listed in the weaknesses section?

Note: I feel the paper is definitely interesting and makes valid contributions. Addressing my questions, in particular the one about the long-term forecasting datasets would be a strong argument for me to raise my score.

Edit: I've read the rebuttal provided by the authors, and since my open questions have been addressed I'm raising my score to 7.

**Limitations:**

Yes

---

> ### Author Rebuttal · Authors · 2024-08-07
>
> We thank the reviewer for comments and suggestions. We address them as follows:
>
> **Aggregate rank metric**
>
> We thank the reviewer for the suggestion. We will add the average rank of each model as:
> | Model                 | Score       |
> |-----------------------|-------------|
> | AutoARIMA             | 21.388889   |
> | Informer              | 14.055556   |
> | Autoformer            | 12.000000   |
> | PatchTST              | 6.833333    |
> | N-HITS                | 11.388889   |
> | MICN                  | 7.277778    |
> | TimesNet              | 11.111111   |
> | LLM-Time              | 12.777778   |
> | TimesFM               | 12.277778   |
> | Lag-LLAMA             | 17.444444   |
> | Chronos               | 13.888889   |
> | STEP                  | 9.111111    |
> | EpiFNP                | 14.666667   |
> | ColaGNN               | 16.500000   |
> | TS2Vec                | 19.111111   |
> | SimMTM                | 15.611111   |
> | TS-TCC                | 18.111111   |
> | LPTM                  | 2.500000    |
> | LPTM-NoSegment        | 12.055556   |
> | LPTM-NoPreTrain       | 10.444444   |
> | LPTM-NoLinProb        | 6.222222    |
> | LPTM-OnlyRandMask     | 7.555556    |
> | LPTM-OnlyLastMask     | 3.666667    |
>
> LPTM has the best average rank of 2.5 with significant lead over second best model.
>
> **Code link**
>
> Sorry. We didn’t realize the link was expired during the review process. The code can be found in the LPTM.zip file at https://osf.io/vy3sc/?view_only=c10508d340ba468287c984496cf57be1
>
> **Long-term forecasting benchmarks**
>
> We have benchmarked on ETT and M4, two popular long-term forecasting benchmarks in the paper. We have added additional benchmarks on ILI, Electricity and Exchange datasets (similar setup to [1]):
>
> | Dataset/Model | AutoARIMA | PatchTST | LLM-Time | TimesFM | Lag-Llama | Chronos | SimMTM | LPTM |
> |---------------|-----------|----------|----------|---------|-----------|---------|--------|------|
> | Electricity   | 0.6       | 0.48     | 0.93     | 0.51    | 0.63      | 0.82    | 0.81   | 0.74 |
> | Exchange      | 1.23      | 0.94     | 1.72     | 0.86    | 1.51      | 1.54    | 0.93   | 0.91 |
> | ILI           | 1.95      | 0.97     | 1.11     | 1.83    | 2.11      | 1.85    | 0.84   | 0.96 |
> | Traffic       | 2.14      | 1.66     | 1.94     | 1.85    | 2.56      | 1.98    | 1.84   | 1.52 |
>
> **Error bars**
>
> We have added the std. Dev across 10 runs in the additional PDF.
>
> ### References
>
> [1] Nie, Yuqi, et al. "A time series is worth 64 words: Long-term forecasting with transformers." arXiv preprint arXiv:2211.14730 (2022).

---

> > ### Comment · Reviewer_RuJ6 · 2024-08-09
> > **Response to rebuttal**
> >
> > I thank the authors for responding to my comments and questions. I feel that the added points strenghten the case for accepting the paper, and have raise the score to 7 as a result.

---

> > > ### Author Response · Authors · 2024-08-09
> > > **Thank you**
> > >
> > > We are glad that our response has addressed your comments and strengthened your case for the paper. We are thankful for the increase in score.
> > >
> > > We would be glad to answer any more of your questions.

---

### Official Review · Reviewer_7ZoZ · 2024-07-12

**Soundness:** 3
**Presentation:** 4
**Contribution:** 3
**Rating:** 6
**Confidence:** 4

**Summary:**

The paper proposes Large Pre-trained Time-series Models (LPTM), a novel method designed to improve the efficiency and performance of time-series analysis across multiple domains.
The key contribution is an adaptive segmentation module that automatically identifies optimal segmentation strategies for diverse datasets during pre-training.
This approach aims to overcome the limitations of fixed-length segmentation, which may not adequately capture the temporal patterns of heterogeneous time-series data.
LPTM demonstrates superior forecasting and classification performance, requiring up to 40% less data and 50% less training time compared to state-of-the-art models.

**Strengths:**

S1. This paper focuses on the time series segmentation problem.
As the basic semantic unit in time series is not as clear as in text, a proper segmentation is a promising direction towards better series modeling.

S2. The proposed segmentation method is adaptively calculated over each specific input series.

S3. The experiments are extensive.

**Weaknesses:**

W1. Although time series has a weaker semantic structure than natural language, it is closer connection to images.
In both time series and images, a semantic unit, e.g., a small item or a texture in an image, can have different lengths and scales.
This raises a challenge against the main motivation: why a full self-attention-based architecture works for images (e.g., ViT), why for time series the segmentation needs to be explicitly done?
It would be interesting if the authors can further discuss this problem and provide their intuitions.

W2. The introduction of the adaptive segmentation module seems to bring instability in the initial model training, as well as requiring longer training time (although the authors propose to backpropagate the gradients every 10 batches).
Specifically, the loss function for segmentation is a hard loss based on the selected subset of best segments.
However, the parameters seem to be randomly initialized, which could provide highly random "best" segments.
Hence, the convergence stability and the training time with and without the dynamic segmentation modules should be discussed.

W3. The dynamic segmentation modules seem not to be fine-tuned with specific attention.
However, as the author(s) mentioned, different datasets could have very different best segmentation.
Hence, it would be interesting to discuss why this is sufficient and provide theoretical or empirical evidences.

**Questions:**

Q1 (cr. W1). Please provide intuitions and empirical evidences on why full self-attention-based architectures work for images while an explicit segmentation module is required for time series.

Q2 (cr. W2). Please discuss the convergence stability, especially the initial convergence stability and the influence of random initialization, as well as the influence of the adaptive segmentation modules on the pre-training speed.

Q3 (cr. W3). Please discuss the influence of fine tuning on the adaptive segmentation, e.g., which the current framework can make a large change to the existing segmentation.
Experiments with and without fine-tuning the adaptive segmentation would be interesting to report.

Q4. It would be interesting if the authors may show some case studies on the adaptive segmentation results, i.e., whether and how much the adaptive segmentation results conform to the source domain of the dataset, whether some periodic patterns can be well preserved after the adaptive segmentation module.

**Limitations:**

L1. There is a lack of explanability and interpretability in the adaptive segmentation results.

---

> ### Author Rebuttal · Authors · 2024-08-07
>
> We thank the reviewer for their comments and we address them as below:
>
> **Intuition on segmentation**
>
> We note that even for vision models such as ConvNets and ViT, the images are ingested as fixed sized patches (such as 16 x 16 patches) which have some similarity to segments in time-series. Moreover, we argue that there is much more  diverse heterogeneity in time-series compared to images since even images of different domains have important properties such as edges, shapes that are useful to capture by the model.
>
> However, for time-series, each domain can have a wide range of patterns specific to the domain along with cross-domain information. Further, these patterns could differ across time. Therefore, segmentation that adapts to these patterns across time can better capture these semantic structures and handle cross-domain downstream tasks.
>
> **Stability of adaptive segmentation**
>
> First, we did not observe instability in training.  We evaluated our model over multiple runs with different random initializations and we didn't experience large variation in stability or performance
>
> Since the overall objective of finding optimal segments as well as pre-train weights of the backbone model on SSL tasks has discrete components we use a bi-level optimization to simultaneously learn both objectives. We update the loss for segmentation every 10 epochs to make the training stable for SSL. This kind of training strategy is used in other bi-level optimizations encountered in discrete tasks such as graph structure learning and reinforcement learning on discrete action space[1,2].
>
>
> **Fine-tuning adaptive segmentation**
>
> We fine-tuned adaptive segmentation to provide optimal segmentation for each domain dataset since the patterns observed across different domains is very heterogenous.
> For example, while we need to capture monthly or yearly patterns for demand datasets, behavioral datasets need to encode different kinds of patterns occurring in order of seconds or minutes.
>
> Training a single segmentation strategy across all domains therefore would be highly suboptimal. In fact, empirically, we observed that training a single segmentation strategy across all domains lead to unstable training due to which we didn’t explicitly include it as a baseline.
>
> **Interpretability of segmentation**
>
> We also show some visualizations of the segmentations for different domains in Appendix Section D. We can get some intuitive understanding from these visualizations such as LPTM capturing smaller segments near epidemic peak to encode fine-grained variations as well as segment lengths correlating with variance of time-series in general.
>
> ### References
>
> [1] Hu, Minyang, et al. "Learning Continuous Graph Structure with Bilevel Programming for Graph Neural Networks." IJCAI. 2022.
>
> [2] Haarnoja, Tuomas, et al. "Soft actor-critic: Off-policy maximum entropy deep reinforcement learning with a stochastic actor." International conference on machine learning. PMLR, 2018.

---

> > ### Author Response · Authors · 2024-08-11
> >
> > We again thank the reviewer for their valuable comments that helped improve our work. We hope we have addressed their questions and concerns.
> >
> > Since the discussion period is soon ending, we hope the reviewer can acknowledge our response. We would also gladly answer any further questions or clarifications.

---

> > > ### Comment · Reviewer_7ZoZ · 2024-08-13
> > >
> > > Thanks to the authors for responding to my concerns. I remain positive about this work, and would like to raise my score to 6.
> > >
> > > Just to clarify a bit about my question Q3, what I would expect are more details on fine-tuning in LPTM. For example, how easily it can adapt to different domains, how quickly it can converge, and how much improvement it can provide. I think this is interesting because I already acknowledged the importance of fine-tuning in dynamic segmentation, and cross-domain is a main topic for this paper. I look forward to the future versions.

---

### Official Review · Reviewer_bKzz · 2024-07-15

**Soundness:** 2
**Presentation:** 2
**Contribution:** 2
**Rating:** 5
**Confidence:** 4

**Summary:**

This paper proposes a novel contribution to pretrained time series models for forecasting and classification by paying attention to the fact that currently several transformer models take time series segmentations of the same size, regardless of the particular characteristics of the time series in consideration. For instance, time series that have yearly frequency or minute frequency might require different segmentation lengths, or it might be that dynamics are more complex in certain time intervals requiring a more detailed segmentation. Based on this observation the authors proposed a model that can find a suitable segmentation schema that later on allows to observe where are the time intervals where more complex dynamics are shown.

The authors perform several experiments and claim empirically that the proposed approach is at least competitive to the state of the art.

**Strengths:**

The authors study a clearly interesting problem: how to provide a suitable segmentation scheme for time series so that different time regions are segmented in different ways, depending on their complexity and amount of information. The motivation for this is well stated by the authors, leading to a novel approach to achieve this.

The authors further set up this in an Self-supervised learning setting, and consider multiple datasets to pretrain their model and further provide several evaluations. This is interesting because depending on the field/topic/area of time series a different segmentation scheme might be more suitable.

**Weaknesses:**

Some of the main limitations are as follows:
- The proposed framework is not differentiable. The authors have acknowledged this in the paper and propose a workaround for this, basically to update the segmentation scores every 10 batches. Yet, this poses challenges like the interpretation of the training loss, and discontinuities in the test loss.
- It is unclear if the proposed approach is able to handle missing values. If not, is there anyway to overcome this? Missing values are very often present in practice and having a sound way to handle them is relevant.
- It is unclear if the current evaluation is fair. The authors present a corpus of datasets for which they pretrained the proposed model, but it is unclear which datasets where hold-out from pretraining. This is relevant as several of the pretrained models considered might have not been exposed to these datasets, which gives an unfair advantage to the proposed model. Further, since the amount of pretraining datasets is rather limited, there is the possibility that the proposed model is overly focused on these datasets, whereas other models, like (Ansari 2024) and (Woo 2024) were trained in a larger corpus of datasets.

**Questions:**

Question:

* Eq-1: is the GRU applied entry-wise to the time series? Does this imply that we apply $GRU_1$ to each entry of $y$ (which has $t$ entries), and then the resulting $t$ values constitute the hidden embeddings?
* Eq. 2: what is $z_i$? So far we have talked about $z^{(i)}$.
* Missing closing parenthesis in fig 1: $S(y^{(1...T)$
* Eq-1: the larger values of S(i,j) the better? Does it mean that the correlation between $z_i$ and $z_j$ is high, or that $z_i$ and $z_j$ are related somehow?
* Eq-3: index $k$ is never used in this definition of output embeddings.
* Eq-5: as pointed out by the authors, the selection of segments is not differentiable and hence it can not be directly integrated to the loss function. Does it mean that the segments are updated every 10 batches? This means that the loss will not be continuous, and hence it will be unclear if there is progress or not in terms of the training loss. Is this correct? I guess here what nevertheless can hint at improvement is the test loss.
* In line 225: why are time series with missing values removed? is the proposed model able to handle missing values?
* The authors claim that their model is a pretrained model. What datasets were used to pretrain the model? Are all datasets used as well for evaluation in Table 1? If yes, then the comparison is not fair. Several of the pretrained models considered might have not been exposed to those datasets in pretraining, giving an unfair advantage to the proposed approach. Further, doing pretraining in such a small amount of datasets further gives more advantage to the proposed model, as the larger the datasets potentially gives a smaller amount of exposure to each dataset.

**Limitations:**

The authors have acknowledged limitations.

---

> ### Author Rebuttal · Authors · 2024-08-07
>
> We thank the reviewer for the comments. We address them as follows:
>
> **The proposed framework is not differentiable..**
>
> As stated in the paper we did not observe any instability in the training.
> Yes, the segmentation of time-series is a discrete operation. LPTM tackles this challenge via a novel differentiable scoring module which learns the best segmentation strategy based on pre-trained dataset.
>
> We note that the frequency of training every 10 batches is key to providing stable training since changing the segmentation module every iteration will change the SSL objective function too frequently leading to instability. Similar strategies are used in many papers that encounter bilevel optimization such as for discrete operations such as graph learning [1] and reinforcement learning over discrete action spaces [2].
>
> **Handling missing values**
>
> Handling missing values is not in the scope of our current work. However, LPTM could be extended to handle missing values using a similar method to random masking task by masking the segments associated with missing values. One issue to tackle would be to selectively mask part of the segment.
>
> **Regarding pre-trained data**
>
> For pre-trainied models such as Lag-LLAMA and TimesFM, we start with the weights of the models publicly available and pre-train on the same datasets we use to pre-train LPTM. Therefore, all baselines are pre-trained with at least as much data as LPTM is to provide a fair comparison. In fact, the other pre-trained models such as TimesFM and Chronos have already been pre-trained on much larger pre-trained datasets. In spite of this, LPTM achieves SOTA performance. For fine-tuning, all models are fine-tined on the same train-test splits.
>
> Further, we use all the datasets described in lines 222-244 for pre-training. Only the unseen test split was used for evaluation on Tables 1 and 2.
>
> **is the GRU applied entry-wise to the time series?**
>
> Yes, for each $y^{i}$ we get an embedding $z^{(i)}$ from the GRU.
>
> **what is z_i?**
>
> This is a typo, we meant $z^{(i)}$.
>
> **Value of s(i,j)**
>
> Yes, larger is the value of $s(i,j)$ more important the segmentation module believes the segment between i and j to be. The intuition for Eq 2 is that since the GRU captures the information at timesteps i and j in context of entire time-series, the segment score function uses this information to derive the segment score.
>
> **Index k in Eq 3**
>
> k is just the indexing variable to aggregate the output embeddings of the self-attention for time-stamps from i to j. We will clarify this.
>
>
> **Regarding Eq 5**
>
> Due to the discrete problem of finding the optimal set of segments, the actual objective function during SSL is not continuous. However, we overcome this by using a bi-level optimization approach and periodically optimize the segmentation by learning on SSL tasks. This provides a more stable approach to training when we encounter discrete objectives similar to such strategies used in many papers that encounter bilevel optimization such as for graph learning [1] and reinforcement learning [2].
>
> **Missing values removed (line 225)**
>
> Many time-series in Project Tycho are very sparse with lots of missing values. We remove these time-series since LPTM and other baselines cannot handle them. However, as noted earlier, extending LPTM to time-series with missing values is an interesting and important challenge for future research.
>
> ### References
>
> [1] Hu, Minyang, et al. "Learning Continuous Graph Structure with Bilevel Programming for Graph Neural Networks." IJCAI. 2022.
>
> [2] Haarnoja, Tuomas, et al. "Soft actor-critic: Off-policy maximum entropy deep reinforcement learning with a stochastic actor." International conference on machine learning. PMLR, 2018.

---

> > ### Author Response · Authors · 2024-08-11
> > **Response to Reviewer**
> >
> > We again thank the reviewer for their valuable comments that helped improve our work. We hope we have addressed their questions and concerns.
> >
> > Since the discussion period is soon ending, we hope the reviewer can acknowledge our response. We would also gladly answer any further questions or clarifications.

---

> > > ### Comment · Reviewer_bKzz · 2024-08-12
> > >
> > > I would like to thank the authors for taking the time for these replies and for addressing my concerns.
> > >
> > > I have updated my rating to 5: Borderline accept.

---

### Author Rebuttal · Authors · 2024-08-07

Tables for the standard deviation of RMSE across 10 runs, mean rank of the models and 3 additional classification tasks.

---

### Decision · Program_Chairs · 2024-09-25

**Decision:**

Accept (poster)

**Comment:**

The submission describes an approach called Large Pre-trained Time-series Models (LPTM) that improves time-series forecasting and classification by introducing an adaptive segmentation module. This module automatically identifies optimal segmentation strategies for different datasets, addressing the limitations of fixed-length segmentation methods. The paper demonstrates that LPTM achieves strong performance across multiple domains with reduced data and training time

The paper received generally positive feedback from reviewers, with a few areas for improvement noted.

**Positive Points:**
* The reviewers appreciated the innovative adaptive segmentation approach, which provides a flexible and effective way to handle diverse time-series data across multiple domains.
* Experiments against a wide range of baselines were seen as a strength.
* The clarity and thoroughness of the paper's presentation, particularly in terms of the problem's relevance and the method's motivation, were commended.

**Negative Points:**
* Several reviewers pointed out the lack of detailed investigation into the interplay between the adaptive segmentation module and the encoder, particularly in understanding why the method works as well as it does.
* Concerns were raised regarding the presentation and reproducibility, including issues with language, formatting, and an expired link to the implementation code.
* The scalability and computational overhead of the adaptive segmentation module were not sufficiently discussed, particularly in relation to the potential increase in tokens and the quadratic scaling of the method.

The LPTM method represents an advance in time-series analysis. While the paper could benefit from a deeper exploration of certain technical aspects and improved clarity in its presentation, these issues do not detract from the overall quality and impact of the work. Given its technical merit and practical relevance, this paper is well-suited for presentation at NeurIPS.